# Indoor mobility challenges among older adults: A systematic review of barriers and limitations

**Xin Er Yaw**[1], **Pei Lee Teh**[2], **Weng Marc Lim**[3], **Shaun Wen Huey Lee**[1]*

1 School of Pharmacy, Monash University Malaysia, Subang Jaya, Selangor, Malaysia, 2 School of Business, Monash University Malaysia, Subang Jaya, Selangor, Malaysia, 3 Sunway Business School, Sunway University, Sunway City, Selangor, Malaysia

* shaun.lee@monash.edu

## Abstract

### Introduction

The global population is projected to double by 2050 with most older adults expressing preference to age in place. Despite this demographic shift, indoor mobility challenges which directly impact independence, safety and quality of life remains poorly understudied, creating a critical knowledge gap for effective intervention development.

### Methods

A systematic literature search was conducted on four databases from inception to September 2024. Inclusion criteria included: (1) participants aged 60 years and above; (2) were primary data on self-reported indoor mobility challenges; and (3) published in English language. Two reviewers independently performed data abstraction, evaluated the risk-of-bias and quality of included article using the NewCastle Ottawa Scale for cross-sectional or Joanna Briggs Institute Checklist for qualitative studies. Thematic synthesis with constant comparison analysis identified key mobility constraint factors.

### Results

Nine studies encompassing 1,833 participants were identified. Older adults reported four main indoor mobility challenges: fatigability, poor balance, home environmental barriers, reduced strength and limb weakness. These limitations contributed to reduced endurance, even in indoor settings. The presence of various health conditions further compromised their mobility and ability to age in place healthily.

### Conclusion

To support ageing in place, a comprehensive environmental assessments and home modifications can be considered. Multi-stakeholder collaborations are needed to

**Data availability statement:** All relevant data are within the manuscript and its Supporting Information files.

**Funding:** The author(s) received no specific funding for this work.

**Competing interests:** The authors have declared that no competing interests exist.

support the independence, safety, and quality of life of older adults who wish to age in place. Further research is warranted in suburban and rural areas to explore socio-economic factors and determinants of indoor mobility.

## Introduction

The global population is experiencing a significant shift, with a notable increase in the population of older adults. By 2030, around one in six people worldwide will be over 60 years old, reaching a total of 1.4 billion individuals [1]. This number is expected to double by 2050, with 2.1 billion older adults, and the number of those over 80 years old will triple to 426 million [2]. This demographic change requires us to understand and address the unique challenges faced by older adults, including age-related health issues, reduced mobility and physical strength, increased risk of chronic diseases, social isolation and limited access to healthcare or support services [2]. These challenges are compounded by the strain on social resources and the shortage of healthcare workforce, highlighting the critical need to ensure that older adults maintain an acceptable quality of life.

Older adults also experience age-related changes that can make daily activities challenging, thereby affecting their independence and overall well-being [3]. Research has shown that grip strength in men and walking speed in women declines with age [4]. Additionally, there is a significant reduction in muscle mass compared to overall body mass as people age, which has broader implications on their daily activities [4]. For instance, muscle atrophy or reduced muscle mass can lead to decreased strength and endurance, affecting tasks like lifting objects, pulling levers, climbing stairs, or engaging in activities requiring muscular power. This, coupled with stiffness, balance issues, and deteriorating vision can make previously simple activities such as climbing stairs or reaching objects a challenge [5–7]. Therefore, addressing these challenges is crucial for maintaining the independence and well-being of older adults, enabling them to age in place successfully [8]. These changes underscore the importance of maintaining an active lifestyle, regular exercise, and strategies to preserve muscle strength and mobility as individuals age. By addressing these challenges, older adults can age in place, reduce the need for institutional care, promote well-being, and prevent falls and injuries [9].

Importantly, for older adults to age in place, it is essential that they are able to navigate their immediate environment and perform essential tasks necessary for independent living. This ability depends on factors such as muscle strength, energy, skeletal stability, joint functionality, and neuromuscular coordination [10,11]. This is in addition to their ability to perform activities of daily living, such as walking, standing, sitting, reaching, stooping, and other relevant movements. Nevertheless, both indoor and outdoor environments pose structural barriers that can impede the mobility of individuals with impairments [12].

Due to the limited mobility, older adults will experience a loss of autonomy, increased fall risk, decreased physical activity, and reduced quality of life. Therefore, it is important to create accessible environments to foster social inclusion and

empower older adults to maintain independence. Despite the importance of this issue, there remains a gap in synthesising existing knowledge and comprehensively understanding this issue. While most older adults typically spend a significant amount of time indoors, most of the studies often focused on their outdoor mobility [13]. Therefore, this review aims to explore the indoor mobility challenges among older adults. Findings of this review can be used by stakeholders such as policymakers, healthcare professionals and caregivers to develop evidence-based policies that address the specific challenges faced by older adults who are age-friendly indoor spaces to promote ageing in place.

## Materials and methods

This study was adhered to the Preferred Reporting Items for Systematic Reviews and Meta-Analyses (PRISMA) statement [14]. The study protocol was registered on PROSPERO (CRD42023438135).

### Search strategy

A comprehensive literature search was conducted to identify for relevant studies addressing indoor mobility challenges among older adults. In this study, indoor mobility was defined as any movement/event that occurred within or inside a residential building or sheltered/covered space such as home, nursing home or residential home. Four electronic databases were searched: PubMed, Embase, CINAHL Plus, and PsycINFO from database inception to September 30 2024. Search terms included indoor mobility, home mobility, indoor activities, home activities, life-space, challenge*, problem*, older adults and geriatric (see S1 File for full search strategy). This was supplemented with a review of references of the included studies to identify any additional relevant articles.

### Study eligibility criteria

We included studies of any design (e.g., cohort, cross-sectional, case reports, qualitative) that examined the indoor space environment and mobility challenges experienced by an older adult aged 60 years and above, published in English language. Studies were excluded if these were: 1) letters to the editor, case reports, commentaries, or abstracts; or 2) were conducted in non-human subjects.

### Study selection, data extraction and risk of bias

References identified from all database searches were imported into Endnote (Clarivate Analytics) and duplicate citations were removed. Two reviewers (XEY, CKN, KFT or PLT) independently screened the titles and abstract for initial eligibility and full-text articles were obtained if considered eligible. Each full-text article was independently assessed for final inclusion. Data extraction was performed independently by two reviewers using a standardised form (S2 File). These data included study characteristics, study location, population characteristics and reported mobility challenges.

Two reviewers independently completed the quality assessment of each study using the NewCastle Ottawa Scale for cross-sectional studies [15] and Joanna Briggs Institute (JBI) Checklist for qualitative studies [16]. Any disagreements between reviewers were resolved through discussion and consensus with input of the last author.

### Data synthesis

Consistent with previous systematic reviews, we summarised results and described them narratively [17,18]. An inductive thematic synthesis was conducted by two authors to analyze the extracted data. This process involved line-by-line coding, theme categorization, and the development of analytical themes, following the approach outlined by Braun and Clarke [19]. Data from various questionnaires were also themed by analysing the underlying meaning and patterns within the numerical data. Results were cross checked by the last author, and discussions were held and cross-validated among reviewers until consensus on the final themes were achieved. Data were analysed using NVivo 12 (Lumivero).

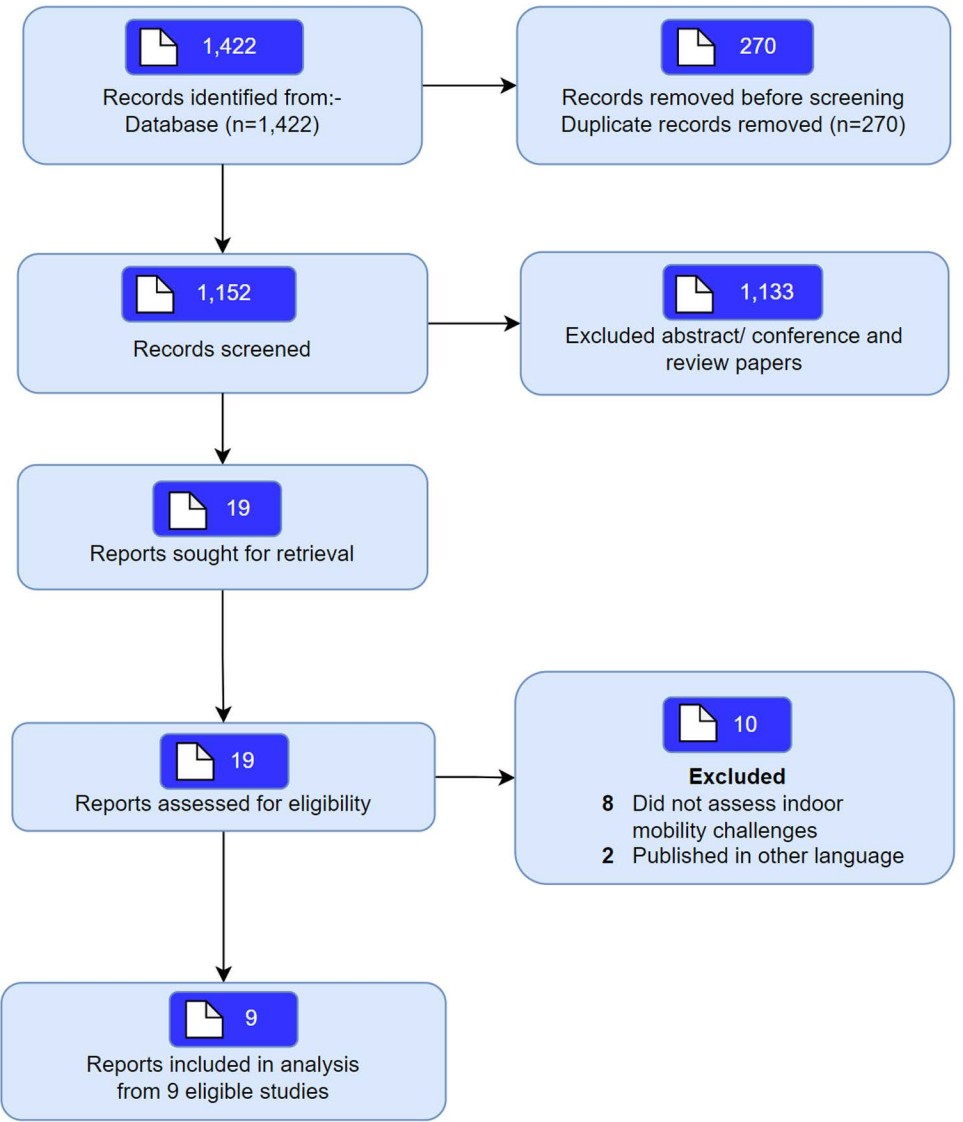

**Fig 1. PRISMA flow chart detailing included/excluded articles.**

## Results and discussion

A total of 1422 articles were identified and 1152 articles were screened for eligibility after removal of duplicates. Nineteen articles were selected for full text screening and nine met the inclusion criteria (Fig 1). Reasons for exclusion included studies that did not assess indoor mobility challenges ($n = 8$) or were published in other languages ($n = 2$, S1 Table). The nine studies recruited a total of 1,833 participants, with the mean age between 69·6 years and 93 years [20,21], and included mostly females. Six studies were performed in Europe, two in United States and one study in Asia, namely Turkey. These studies were conducted in either a community setting [20–27], institutionalised setting [21,23–25,28] or both. The characteristics of the included studies are found in Table 1. The studies included in this review were rated as having medium to high methodological quality (S2 Table and S3 Table). Most of the included studies had a satisfactory level of

**Table 1. Characteristics of included studies.**

| Author (year) Location | Study characteristics | Assessed variables | Key Findings | Limitations/Strengths | Conclusion |
|---|---|---|---|---|---|
| Brim B (2021) USA | Retrospective review Semi-structured interview (Qualitative) (n = 36) Age range: > 55 years Female: Unspecified | •Barriers to ageing in place were identified through home safety assessment reports | •Several overarching categories of barriers to ageing in place were identified, including home mobility and safety, home improvement and maintenance needs. •Older adults can recognise many barriers within their homes that hinder ageing in place. •Accessing support and services to overcome these barriers requires additional resources and funding, indicating a need for further investment in this area. | Strengths: •Grounded theory approach provides valuable insights into ageing in place from older adults' perspectives. •Multiple interviewers: Increased trustworthiness of reports Limitations: •Retrospective design: potential recall bias and incomplete information. •Not originally designed as a research study, potentially compromising methodological rigour. •Lack of demographic data limits generalisability. •Unavailability of verbatim statements reduces nuance. •Potential sampling bias from participants requesting home safety assessments. •Small sample size limits generalisability and statistical power. •Limited geographic scope may not capture the experiences of suburban or rural older adults. | This study emphasises the knowledge gap and barriers older adults face in accessing resources for ageing in place. Comprehensive ageing programs and resources, including assessment and preventive services by occupational therapists, are essential. Future funding and health insurance reimbursement is crucial for ageing-in-place initiatives. |
| Brustio PR (2022) Italy | Cross-sectional: Physical assessment (Quantitative) (n = 32) Age range: 65–84 years Female: 56·2% | •Robust and frail participants were compared •Independent variable: Robust or frail status •Dependent variable: Indoor mobility, Frailty and disability | •Robust participants had higher indoor mobility (IM) and lower disability compared to frail participants. •Physical frailty mediated the relationship between IM and disability, indicating reductions in IM may contribute to increased physical frailty and subsequent disability. •Total frailty score did not mediate the relationship between IM and disability. •Frail participants showed greater restriction in IM and higher levels of disability. | Limitations: •Limited generalisability: sample consists purely of volunteers. •Cultural context may influence results, limiting generalisability. •Cross-sectional design prevents establishing causal relationships. •Potential influence of seasonal variation on activity levels, affecting generalisability. Strengths: •Examined important variables (IM, frailty, disability) in older adults, providing valuable insights into health status. •Included relatively healthy community-dwelling older adults capable of independent mobility and daily activities. •Consistent with previous research, supporting the association between IM, frailty, and disability. •Conducted mediation analysis to explore the role of physical frailty as a mediator. | Frail participants showed greater restriction in IM and exhibited higher levels of disability. IM reduction may have a negative impact on physical frailty and indirectly increase disability. |

**Table 1.** (Continued)

| Author (year) Location | Study characteristics | Assessed variables | Key Findings | Limitations/Strengths | Conclusion |
|---|---|---|---|---|---|
| Clemencon M (2008) France | Cross-sectional: Descriptive observational Quantitative: Measurements (maximal leg power, optimal velocity, optimal torque), Physical performance tests (walking speed test, chair rise test, stair climb test) (n = 39) Age range: 72–96 years Female: 100% | •Maximal leg power, optimal velocity, and optimal torque, were compared to determine their association with physical performance measures, e.g., 6 m walking speed, chair-stand time, and stair-climb time. •Independent variables: Maximal leg power, optimal velocity, optimal torque •Dependent variables: Physical performance measures (6 m walking speed, chair-stand time, stair-climb time) | •Maximal leg power and optimal velocity significantly correlate with physical performance measures. Hence, they are identified as determinants of physical performance and are considered significant mobility factors •Optimal torque does not correlate with physical performance measures. •Velocity-oriented training may be valuable in improving functional status. | Limitations: •Lack of assessment of leg asymmetry •Absence of surface electromyography (EMG) to evaluate neural activation and muscle coordination during physical performance. Strengths: •Objective measurement of variables enhances data reliability and accuracy. •Multiple outcome measures to assess physical performance. •Use of standardised tests to ensure consistency and comparability of results. | Maximal power and optimal velocity are important predictors of physical performance in older women. Muscle mass loss is not the primary factor affecting performance, but rather type II fibre percentage and activation capabilities. These findings suggest the need for training programs that prioritise contraction velocity to enhance muscle power and delay functional decline. Future research should explore optimal velocity training protocols and their impact on muscle function and mobility. |
| Cress ME (2011) USA | Cross-sectional: Physical assessment, Living space area (Quantitative) (n = 61) Age range: 65–94 years Female: 60 · 66% | •Community dwellers (CD) and residents of a retirement community (RC) were compared. •Independent variable: Living conditions (CD vs RC) •Dependent variable: Living space area, physical activity and function | RC residents had smaller living spaces and took fewer total steps per day (excluding exercise) compared to CD participants. The RC group also exhibited lower physical function. | Limitations: •Limited focus on home demands, neglecting factors, e.g., caregiving responsibilities or non-step-related activities. •Sample bias due to volunteer participants, predominantly Caucasian and highly educated, limiting generalisability. •Limited applicability: excluded individuals with cognitive impairments or severe functional limitations. •Potential influence of seasonal variation on activity levels, affecting generalisability. Strengths: •Clear association found between home size and physical activity, measured by daily steps. •Use of diverse assessment methods for comprehensive data collection. •Analysis considering covariates, e.g., age and health factors. | Home size and physical function were primary predictors of the number of steps taken at home. |

*(Continued)*

| Author (year) Location | Study characteristics | Assessed variables | Key Findings | Limitations/Strengths | Conclusion |
|---|---|---|---|---|---|
| Geren AA (2006) Turkey | Cross-sectional: Physical performance tests (Quantitative) (n = 64) Mean age: 72·5 years (Institutionalised); 69·6 years (Home-dwelling) Female: 28% | •Older adults living at home and institution were compared •Independent variable: Living condition (home vs. institution) •Dependent variable: Physical performance measures, i.e., mobility, balance, and physical performance | Older adults living at home had better physical performance than those living in an institution, as evidenced by the 6-minute walk test. There were no significant differences in demographic and physical characteristics between the two groups. Age and mobility were correlated with physical performance in both groups, indicating that older age and lower mobility negatively impact physical performance. Institutional living may adversely affect mobility, balance, and overall physical performance in the elderly. | Limitations: •Potential confounding factors: Other unaccounted confounding factors could influence results, e.g., individual health conditions, medication use, and socioeconomic status. •Longitudinal data would provide more insights into the changes in mobility, balance, and physical performance over time and their association with living arrangements. Strengths: •Direct comparison of mobility, balance, and physical performance between older adults living at home and institutions to assess differences. •Investigated correlations between physical performance, age, and mobility in both groups. | Older adults who live in institutions had lower mobility, balance and physical performance than those who live at home. Age and lower mobility negatively influence the physical performance of older adults. |
| Mänty M (2012) Denmark | Cross-sectional: Face-to-face interview to collect mobility related fatigability and other health factors. (Quantitative) (n = 1181) Age range: 92–93 years Female: 71·28% | •Sheltered housing participants and independent living participants were compared •Independent variable: Cardiovascular diseases, musculoskeletal pain, medications, walking speed, and depressive symptoms •Dependent variable: Indoor-mobility-related fatigability | •Prevalence of Fatigability: Approximately one-fourth (26%) of participants reported experiencing fatigue when transferring or walking indoors. •Fatigability was more prevalent in participants living in sheltered housing (32%) compared to those living independently (23%). •Associated Health Factors: Cardiovascular diseases, musculoskeletal pain, medications, walking speed, and depressive symptoms may contribute to fatigability in nonagenarians. | Limitations: •Cross-sectional design limits the ability to establish causality between fatigability and associated health factors. •Reliance on self-report measures: Susceptible to recall bias or social desirability bias. •Lack of information on other potential factors, e.g, social support, cognitive function, or lifestyle factors. Limits comprehensiveness of the analysis. •Limited generalisability to other populations: Findings may not be directly applicable to younger age groups or populations from different cultural backgrounds. Strengths: •Well-characterised nationwide cohort increases generalisability. •Standardised measures ensure consistency and reliability in data collection, allowing for more robust and valid results. •Inclusion of multiple health-related factors provides broader understanding of associations with fatigability. | Fatigability in very basic everyday mobility is relatively common in non-disabled nonagenarians. Results indicate important associations between fatigability and potentially modifiable health factors. |

*(Continued)*

Table 1. (Continued)

| Author (year) Location | Study characteristics | Assessed variables | Key Findings | Limitations/Strengths | Conclusion |
|---|---|---|---|---|---|
| Martin P (2021) Germany | Cross-sectional: Phone interview to collect data for questionnaires (Quantitative) (n = 222) Median age (years): Nursing home Female: 86 Male: 83 Independent Female: 78 Male: 76.5 Female: 72 · 97% | •Nursing home and independent living participants were compared •Independent variable: Living arrangements (Nursing home vs independent living) •Dependent variable: Self-assessment and external assessment of mobility, quality of life, activities of daily living (ADL). Scores obtained from clinical scoring systems used to assess mobility, e.g., Parker Mobility Score, Barthel Index, and EQ-5D-5L. | •Participants overestimate their mobility, quality of life, and activities of daily living (ADL). •Females in nursing homes show significantly higher self-assessment scores for Parker Mobility Score and Barthel Index. •Both females and males in nursing homes and those living at home have significantly higher self-assessment EuroQoL-5 dimension (EQ-5D-5L) scores. •Older adults, particularly those in nursing homes, tend to overestimate their mobility and functional abilities. | Limitations: •Heterogeneity and subjectivity of external assessors may lead to variability and biased assessment of mobility. •Relying solely on self-assessment and verbal statements may not provide an accurate understanding of patients' mobility due to potential discrepancies and influence from emotional dependency or mental condition. •Findings may be representative of settings with large clinics and specialised orthogeriatric care, but generalizability to other settings is limited. Strengths: •The Parker Mobility Score is recommended for assessing mobility in most groups. •The Barthel Index is suitable for evaluating ADL, especially for independent individuals. •The EQ-5D-5L subjectively confirms a high quality of life. •Slight cognitive impairment does not significantly affect self-assessment accuracy. •Objective geriatric assessment with standardised tests complements self-assessment. •Wearables provide objective data for personalised therapy and rehabilitation outcomes in orthogeriatric patients. | Individuals over 65 years tend to overestimate their level of mobility, quality of life, and ADL. Especially in nursing home participants. Caution is needed when using scoring systems for assessing older adults in nursing homes, due to discrepancies between self and external assessment. |

*(Continued)*

**Table 1.** (Continued)

| Author (year) Location | Study characteristics | Assessed variables | Key Findings | Limitations/Strengths | Conclusion |
|---|---|---|---|---|---|
| May D (1985) UK | Cross-sectional Questionnaire, gait and balance tests, life-space diary (Quantitative) (n = 24) Age range: 64–88 years Female: 70% | •Life-space diameter was compared with mobility index, gait and balance •Independent variable: Life-space diameter •Dependent variable: Mobility index scores, gait and balance tests | •Medical reasons for not going out: leg pain, stiffness, breathlessness, weakness, fatigue, fear of falling, dizzy attacks, chest pain, claudication, poor eyesight. •Non-medical factors for not going out: caring for a sick relative, fear of burglary, lack of purpose or companionship, adverse weather conditions. •Distribution of mobility index scores showed two sub-groups: relatively fit and less fit subjects. •Higher gait speeds correlated with higher mobility index scores. •Better balance was associated with higher mobility index scores, as measured by mean sway path. •Participants mentioning "fear of falling" had lower mobility index scores, slower gait speeds, and higher mean sway paths. | Limitations: •Study focuses on mobility in the home environment and does not address the broader aspects of mobility, such as community participation or access to essential services. •Other gait parameters measured were highly correlated with gait speed but provided little additional information. Analysis may lack comprehensive measures of mobility and potential contributing factors beyond gait speed and balance. •Study does not explore reasons behind the fear of falling expressed by individuals with restricted outdoor mobility and impaired balance, leaving potential underlying causes unexamined. Strengths: •Captures actual mobility achieved by the participants, instead of relying on subjective assessments of capabilities. •Demonstrates a close relationship between mobility, gait speed, and balance, suggesting a strong association between these factors. •Participants with good gait speed and balance tended to go out frequently, despite other symptoms that might restrict their mobility, providing valuable insights into the importance of these factors in determining mobility. | Life-space diary is useful in recording mobility and evaluating therapy effectiveness in impaired mobility conditions. Understanding elderly mobility requires distinguishing between different housebound statuses and types of mobility. Good balance and fast gait speed were linked to increased mobility. Gait speed was strongly correlated with other parameters. Older individuals walking > 1 m/s have good balance and satisfactory mobility, while those <0·5 m/s experience restricted mobility, impaired balance, and fear of falling. |

*(Continued)*

**Table 1.** (Continued)

| Author (year) Location | Study characteristics | Assessed variables | Key Findings | Limitations/Strengths | Conclusion |
|---|---|---|---|---|---|
| Tsai LT (2014) Finland | Cross-sectional: Physical activity measurement, Life-space mobility assessment (Quantitative) (n = 174) Age range: 75–90 years Female: 63 · 54% | •Older adults with restricted and unrestricted life-space mobility were compared •Independent variable: Restricted or unrestricted life-space mobility •Dependent variable: Objectively measured physical activity | Positive correlation between higher physical activity levels and greater life-space mobility. Approximately 16% of participants had a restricted life-space area and were found to be less physically active compared to those with a broader life-space area. Among this 16% of participants, around 70% had exceptionally low daily step counts (≤ 615 steps) and moderate activity time (≤ 6·8 minutes). | Strengths: •Novelty: First study to examine the association between objectively measured physical activity and life-space mobility. •Objective measurement: Utilised 7-day accelerometer data for accurate assessment of physical activity. •Generalisability: Diversity in age and gender, enhancing the applicability to independent older adults in the community. Limitations: •Underestimation of physical activity due to not considering activity bouts and certain activities (e.g., cycling or swimming) •Uncertainty regarding accelerometer sensitivity for older adults using assistive devices •Reliance on self-reported variables introduce potential reporting bias •Lack of adjustment for assistive devices may affect associations between physical activity and life-space mobility •Inability to establish causal relationships due to cross-sectional design •Need for more detailed studies to explore specific travel destinations for intervention purposes. | Prospective studies are needed to determine the temporal order between low physical activity levels and the restriction of life-space mobility. It is necessary to investigate whether low physical activity leads to a restricted life space or if a restricted life space contributes to lower physical activity levels. |

methodological rigour. However, there were some limitations in the study design which may limit the generalisability of the results [20–23,25].

## Summary of findings – Indoor mobility challenges

Factors contributing to indoor mobility challenges faced by older adults were organised into four main themes as below (Fig 2).

### Fatigability

Activity tolerance in older adults encompasses the ability to engage in various activities and maintain different body positions, including walking, climbing stairs, and performing kitchen or cleaning tasks. Older adults commonly attribute difficulties in these activities to self-reported musculoskeletal or cardiopulmonary conditions [22]. Two studies also reported how indoor mobility activities of older adults is limited due to reduced endurance or fatigability, leading to slower walking and gait speeds [21,28]. Indeed, nearly one in every four older adult (22.9%) were reported to experience fatigue when transferring or moving indoor, with 32.1% reported fatigue especially among those with co-morbidities [21]. In another study by Clemencon and colleagues, the authors that leg muscle mass can be an important parameter to use as it predicts fatigability and therefore physical performance.

As individuals age, the prevalence of chronic diseases rises, with more than half of the population (68.8%) requiring pharmacotherapies [20]. The use of medication was found to impact their mobility as older adults reported experiencing fatigue with these medication use. In particular, there was a 10% increase in the likelihood of reporting indoor mobility-related fatigue with every additional medication taken by an older adult [21]. Additionally, older adults reporting musculoskeletal pain were twice as likely to report fatigue, and the presence of depressive symptoms was significantly associated

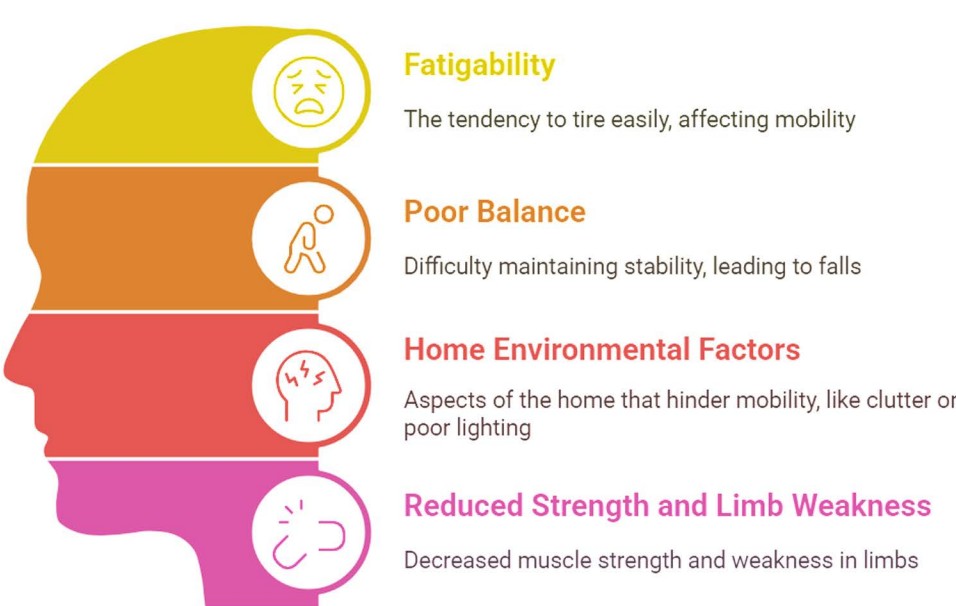

Fig 2. Indoor mobility challenges of older adults.

with reported fatigability, which led to reduced mobility [21]. Among frail older adults, this fatigue led to decreased step counts, increased sedentary behaviour, and greater levels of disability compared to their more robust cohort [20].

### Poor balance

Functional mobility and balance, stemming from previous falls, are common health barriers among older adults, leading to decreased balance, unsteadiness, and concerns about night time bathroom accessibility [15]. While home-dwelling elders generally exhibit better balance compared to institutionalised elderly individuals, approximately one-third of older adults expressed fear of falling, significantly impeding their mobility [17]. May and colleagues established an inverse relationship between balance (measured by mean sway path) and mobility index, indicating that better balance corresponds to higher mobility scores. Conversely, individuals with a fear of falling display lower mobility, slower walking speeds, and higher mean sway path values, indicating poorer balance [19]. While reasons for the poor balance vary, these were identified due to several factors, including the use of medications, the consumption of alcohol, inner ear imbalance as well as certain medical conditions, such as cognitive impairment [19]. Importantly, for an older adult to be mobile out of doors, this required them to have a moderate or rapid gait speed, which are often lacking among those with poor balance [26].

The review also noted an association between cognitive impairment and the likelihood of misjudging daily activities, especially those living in nursing homes [25]. These results underscore the importance of considering cognitive state when evaluating self-assessed mobility, especially for those living in nursing homes [25]. Mobility limitations negatively impacted daily activities including feeding, bathing, grooming, dressing, continence (bowels and bladder), toilet use, transfer, mobility, and stairs climbing [25]. Studies also reported that due to hearing loss, bowel and bladder issues, and vision impairments contribute to poor balance, thereby increasing the risk of falls [26].

### Home environment

The home environment was identified as the key factor that can either support or hinder the indoor mobility of older adults, often contributing to falls due to poor balance. In particular, older adults described the importance of creating a safe and accessible living environment at home to support indoor mobility among older adults [22]. For instance, hazards such as damaged or broken stairs and railings, poor lighting, loose areas of rugs and obstacles in walkways were reported to cause these older adults to be wary, as they worry about the risk of falling [22]. Due to these factors, older adults were reported to limit their movements indoors which led to reduced mobility indoors and decline in independence [26].

### Reduced strength and limb weakness

Both reduced physical strength and limb weaknesses due to ageing among older adults were reported to significantly impact the ability of older adults to move indoors, with a concomitant toll on access to daily activities. For instance, studies have reported that older adults experience a reduced range of motion, which limits their mobility, flexibility, and walking speed [23]. These limitations were more pronounced among those living in retirement communities, as they were reported to engage in fewer daily steps, and exhibit lower activity levels [20,23,27] compared to those living independently in the community [24]. This limitation was even more pronounced among those with frailty, where Brustio and colleagues reported that among frail older adults, they were more likely to experience indoor mobility restrictions and higher levels of disability [20].

In this study, we aimed to summarise the indoor mobility challenges experienced among older adults. The findings suggest that physical limitations, such as reduced physical capacity, poor balance and coordination, and decreased endurance can significantly impact indoor mobility among older adults [13,29,30]. In particular, for frail older adults, this is an important mediating factor that further affects autonomy and disability [31], suggesting the potential benefits of physical activity interventions such as strength training, balance exercises and gait training for this population [31,32].

The study findings provide support for previous research, indicating a significant relationship between chronic diseases and medication use, as well as cognitive and depression with indoor mobility challenges [33–35]. For instance, studies have reported that the use of Z-drugs and anticholinergics can impair functional status among older adults [36,37]. These highlight the important relationships between suboptimal prescribing, and functional status which can impact mobility. As such, one effective intervention could involve the deprescribing of medication, or withdrawal of medications [38]. Studies to date have consistently reported the benefits of deprescribing in reducing the number of falls, which can potentially improve mobility among older adults [39].

The potential of environmental assessments and structural home modifications is another important aspect to consider in promoting indoor mobility and fall prevention among older adults, especially among older adults dwelling in the community. Studies have shown that environmental modification such as installation of grip bars in the bathroom can potentially reduce falls. The consequences of falling are detrimental for older adults as they are susceptible to fractures [40]. A practical approach would be to offer environmental assessments and modifications to older adults who have recently experienced a fall. This process should include a comprehensive physical assessment, such as the home fall prevention checklist for older adults by Center for Disease Control and Prevention [41], providing home modification to improve task performance and safety and use of assistive technology such as grab rail [42], in-home monitoring and domestic robots [43,44] to maintain or improve independence in this group [45,46]. Additionally, home design changes, such as installing lower-height cupboards and switches, can accommodate older adults with reduced range of motion, making it easier for them to perform tasks such as placing items in overhead cupboards or turning on switches. It is also important to consider the preferences and abilities of older adults in the implementation of these modifications to ensure their maximum success.

## Areas requiring further investigation

**Geographic scope and challenges in suburban and rural areas.** Expanding research beyond urban areas to include suburban and rural settings is important. Investigating indoor mobility challenges in these regions will uncover specific environmental, social, and infrastructural factors that impact mobility. Tailored interventions can then be developed to address the distinct needs of older adults in different geographic contexts. For example, in the United States, Continuing Care Retirement Community (CCRC) provides a form of ageing in place in which an individual can transition between different units within the same community as their needs change [47]. Another example is the Certified Aging-in-Place Specialist (CAPS) program that is developed to support professionals who are engaged in making structural changes to improve the safety of older adults' homes [48].

**Socioeconomic factors and determinants.** To date, there is limited understanding of how socioeconomic factors affect indoor mobility among older adults. As such further exploration and understanding how socioeconomic status (e.g., income, education, and housing affordability) affects indoor mobility will help identify specific barriers and facilitate the development of strategies that promote equitable access to resources. Understanding how socioeconomic factors shape the physical environment, access to assistive devices, social support networks, caregiving arrangements, and healthcare utilisation is vital for developing interventions that address the underlying determinants of indoor mobility challenges. From the perspective of the Selection, Optimization, and Compensation (SOC) model [49] of successful ageing, older adults who set goals to maximise their mobility capabilities are more likely to experience a better quality of life. Implementing multiple strategies to address mobility challenges, such as optimising available resources and compensating for physical limitations, could empower older adults to maintain their independence and improve their overall well-being.

**Living environment.** Our review also suggests that there may be baseline differences in terms of frailty and cognitive ability of older adults living in the community and institutional settings [23,26]. As such, different levels of interventions may need to be tailored for these individuals from different setting. For instance, as older adults living in institutional settings are more likely to be frail, they may require interventions focusing on compensatory strategies compared to

those living in the community which may require exercises to improve their strength and balance. Future research should explicitly separate and compare indoor mobility challenges between institutional and community settings to develop more targeted and supportive care approaches.

**Implications and applications.** Given the multifaceted nature of ageing and mobility, there is a need for collaboration among various stakeholders. Healthcare professionals play a vital role promoting mobility in older adults. For instance, they could encourage older adults to participate in strength training, balance exercises, and gait training, which have been found to be effective in promoting physical strength [32,50]. In addition, physicians and pharmacists should regularly conduct medication review to optimise the medication regimens and minimise side effects, while also educating older adults about the potential impact of medications on mobility [34,51,52]. Collaboration between healthcare professionals, pharmacists, and older adults can ensure appropriate medication use and minimise its impact on mobility.

Another aspect that could be included is the use of cognitive assessments into mobility evaluations in nursing homes to identify individuals at risk of misjudging their abilities [25]. Accessible mental health services should be readily available to address depressive symptoms and provide support for cognitive changes [53]. Caregivers and family members should be educated about the relationship between cognitive impairment and mobility to provide appropriate assistance and support. Community organisations and support groups should offer resources and programs that promote cognitive health, social engagement, and mental well-being among older adults [54].

Housing providers and policymakers should prioritise the implementation of accessible design principles, including adequate lighting, handrails, non-slip surfaces, and the removal of environmental hazards [55]. Collaboration between ageing-in-place programs and community services is essential to provide necessary resources such as donation services, exercise programs, physical therapy referrals, and transportation services [56]. Accessible and affordable home improvement and repair services should be made available to address maintenance needs and ensure home safety [57]. For residence facilities, Guidelines for Design and Construction of Residential Health, Care and Support Facilities developed by Facility Guidelines Institute could be useful [58].

## Strengths and limitations

The study offers several strengths. Firstly, this study offers a comprehensive review of the literature by integrating both qualitative and quantitative data, providing a holistic understanding of the indoor mobility challenges faced by older adults. We also identified several research gaps and offered future directions, which can serve as a valuable resource for policymakers, healthcare professionals, and stakeholders, informing interventions and evidence-based practices.

However, limitations and potential biases must be acknowledged. Language restrictions may have limited the inclusion of studies from diverse cultural and socioeconomic contexts. The absence of quantitative data synthesis hinders precise effect estimates and strength of evidence assessment, reducing overall robustness. The subjective nature of narrative reviews poses a risk of over-generalisation or sweeping statements based on limited evidence. Despite these limitations, the integration of qualitative and quantitative data contributes to a broader understanding of indoor mobility challenges in older adults. Future research should address these limitations by including more qualitative studies, conducting quantitative synthesis where appropriate, and adopting rigorous methodologies to strengthen the evidence base.

## Conclusions

In summary, this study identified challenges older adults face in indoor mobility, including physical limitations, frailty, chronic diseases, medication use, and environmental barriers. To optimize implementation, future studies should examine the cost-effectiveness of different implementation models across diverse geographic and socioeconomic contexts to create more adaptable approaches to aging-in-place. Our findings also challenge the current paradigm that aging-in-place can succeed without systematic support structures. The evidence demands not just acknowledgment of these challenges but decisive action to transform how we support indoor mobility among older adults.

## Supporting information

**S1 File: Search strings used in the current study.**
(DOCX)

**S2 File: Data extraction form for the systematic review.**
(DOCX)

**S3 File: PRISMA 2020 Checklist.**
(DOCX)

**S1 Table: List of excluded studies with reasons.**
(DOCX)

**S2 Table. Newcastle-Ottawa Scale (NOS) Assessment Tool.**
(DOCX)

**S3 Table. Joanna Briggs Institute (JBI) Assessment Tool.**
(DOCX)

## Author contributions

**Conceptualization:** Pei Lee Teh, Weng Marc Lim, Shaun Wen Huey Lee.

**Data curation:** Xin Er Yaw, Shaun Wen Huey Lee.

**Formal analysis:** Xin Er Yaw, Shaun Wen Huey Lee.

**Funding acquisition:** Pei Lee Teh.

**Investigation:** Xin Er Yaw, Weng Marc Lim.

**Methodology:** Shaun Wen Huey Lee.

**Project administration:** Pei Lee Teh, Shaun Wen Huey Lee.

**Writing – original draft:** Xin Er Yaw, Shaun Wen Huey Lee.

**Writing – review & editing:** Pei Lee Teh, Weng Marc Lim, Shaun Wen Huey Lee.

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
