## [Decision Letter · Decision Letter 0]

5 Mar 2025

PONE-D-24-49704Mobility Challenges of Navigating Indoor Spaces Among Older Adults: A systematic reviewPLOS ONE

Dear Dr. Lee,

Thank you for submitting your manuscript to PLOS ONE. After careful consideration, we feel that it has merit but does not fully meet PLOS ONE’s publication criteria as it currently stands. Therefore, we invite you to submit a revised version of the manuscript that addresses the points raised during the review process.

We look forward to receiving your revised manuscript.

Kind regards,

Jan Chrusciel

Academic Editor

PLOS ONE

3. As required by our policy on Data Availability, please ensure your manuscript or supplementary information includes the following:

Reviewers' comments:

Reviewer's Responses to Questions

**Comments to the Author**

1. Is the manuscript technically sound, and do the data support the conclusions?

Reviewer #1: Yes

Reviewer #2: Partly

2. Has the statistical analysis been performed appropriately and rigorously? 

Reviewer #1: N/A

Reviewer #2: N/A

3. Have the authors made all data underlying the findings in their manuscript fully available?

Reviewer #1: Yes

Reviewer #2: Yes

4. Is the manuscript presented in an intelligible fashion and written in standard English?

Reviewer #1: Yes

Reviewer #2: Yes

5. Review Comments to the Author

Reviewer #1: This systematic review presents a well-structured and informative synthesis of indoor mobility challenges among older adults. The study effectively outlines its objectives, methods, and findings, with a clear adherence to PRISMA guidelines and a well-defined search strategy. The thematic synthesis is appropriate, and the risk-of-bias assessment strengthens methodological rigor.

However, minor grammatical inconsistencies, structural redundancies, and inconsistencies in terminology impact readability. Below are the key areas for refinement:

1. Title Revision for Clarity & Precision:

Suggested: "Indoor Mobility Challenges Among Older Adults: A Systematic Review of Barriers and Limitations."

2. Abstract Refinements:

o Revise redundant phrases (e.g., "worsening of health issues due to age-related declines in physical strength and mobility").

o Improve clarity in methodology description:

"Two reviewers independently conducted data abstraction, risk-of-bias assessment, and quality evaluation."

3. Introduction:

o Minor grammatical refinements (e.g., "Research have shown" → "Research has shown").

o Ensure terminology consistency by using "older adults" uniformly.

4. Methodology Refinements:

o Revise: "Data analysis was performed using NVivo 12 (Lumivero)."

Suggested: "Data were analyzed using NVivo 12 (Lumivero)."

o Improve consistency in citation formatting and refine descriptions for readability.

5. Results Section Refinements:

o Sentence restructuring for clarity:

"Mobility limitations negatively impacted daily activities, including feeding, bathing, grooming, dressing, continence (bowel and bladder), toilet use, transfers, general mobility, and stair climbing."

o Ensure correct reference formatting (e.g., "Appendix Tables 2 and 3").

o Improve thematic descriptions for readability and flow.

6. Discussion & Conclusion:

o The conclusion would benefit from a stronger emphasis on the implications for policy and practice.

o Revise: "A comprehensive environmental assessments and home modifications can be considered."

Suggested: "Comprehensive environmental assessment and home modifications should be considered to support ageing in place."

Reviewer #2: Thank you for asking me to review this article which is interesting but required further analysis. Below are my comments.

The search strategy is not included in the appendix of the study.

The authors use NewCastle Ottawa Scale and Joanna Briggs Institute checklist. Since some included studies were mixed method how was this dealt with? Why a MM checklist not used instead?

The data synthesis included a narrative however, no descriptive data on the quantitative studies is included. Unclear how the thematic analysis from all the studies was used for all the data together.

The results need to include actual numbers not mostly or majority.

The main themes are presented without a general description of the theme and no supporting results such as quote, or actual numbers are presented for the themes.

In the discussion some of the ideas discussed are not related to the results presented previously. There is no integration of the qualitative and quantitative results. The discussion is generics.

Examples need to consider socioeconomic factors was this data not available or not extracted from the articles.

Conceptually it is unclear how long-term care institutions and community houses were grouped together when the cohort of older people is different. Also, long-term care institutions are usually licences and have minimum environmental standards to people who are frail while personal houses do not. Presenting the data jointly limits the analysis and possible recommendations.

6. PLOS authors have the option to publish the peer review history of their article (what does this mean? ). If published, this will include your full peer review and any attached files.

**Do you want your identity to be public for this peer review?** For information about this choice, including consent withdrawal, please see our Privacy Policy .

Reviewer #1: No

Reviewer #2: No

---

## [Author Response · Author response to Decision Letter 0]

11 Mar 2025

10 May 2025

Jan Chrusciel

Academic Editor

PLOS ONE

Dear Dr Jan Chrusciel,

Thank you very much for kindly considering our manuscript entitled “Indoor mobility challenges among older adults: A systematic review of barriers and limitations” for submission to PLOS ONE. As requested by the reviewer, we have made substantial changes to our manuscript. These changes are delineated as follow:-

The editor had the following comments

Comment #1 Please ensure that your manuscript meets PLOS ONE's style requirements, including those for file naming. The PLOS ONE style templates can be found at

We take note and edited our manuscript to conform with PLOS ONE’s style requirements.

Comment #2. We note that the grant information you provided in the ‘Funding Information’ and ‘Financial Disclosure’ sections do not match. When you resubmit, please ensure that you provide the correct grant numbers for the awards you received for your study in the ‘Funding Information’ section.

We take note and edited it to ensure it matches

Comment #3. As required by our policy on Data Availability, please ensure your manuscript or supplementary information includes the following: A numbered table of all studies identified in the literature search, including those that were excluded from the analyses.

We take note and edited our supplementary information as required as S1 Table

Comment #4. For every excluded study, the table should list the reason(s) for exclusion.

We take note and edited our supplementary information as required as S1 Table.

Comment #5. If any of the included studies are unpublished, include a link (URL) to the primary source or detailed information about how the content can be accessed.

We take note and have ensured compliance. All our included articles have been published in journals and therefore can be easily accessed.

Comment #6. A table of all data extracted from the primary research sources for the systematic review and/or meta-analysis. The table must include the following information for each study:

• Name of data extractors and date of data extraction.

We take note of these requirements. We have included a copy of our data extraction table used in our study into the supplementary information.

Reviewer #1 had the following comments.

Comment #1. This systematic review presents a well-structured and informative synthesis of indoor mobility challenges among older adults. The study effectively outlines its objectives, methods, and findings, with a clear adherence to PRISMA guidelines and a well-defined search strategy. The thematic synthesis is appropriate, and the risk-of-bias assessment strengthens methodological rigor. However, minor grammatical inconsistencies, structural redundancies, and inconsistencies in terminology impact readability. Below are the key areas for refinement

We thank the reviewer for the encouraging comments and suggestions. We have edited the manuscript according to the suggestions of the reviewer.

Comment #2. 1. Title Revision for Clarity & Precision: Suggested: " Indoor Mobility Challenges Among Older Adults: A Systematic Review of Barriers and Limitations."

We thank the reviewer for the suggestion. We have now edited our manuscript title as suggested to read as follow:-

Indoor Mobility Challenges Among Older Adults: A Systematic Review of Barriers and Limitations

Comment #3. Revise redundant phrases (e.g., "worsening of health issues due to age-related declines in physical strength and mobility").

We take note and thank the reviewer for the suggestion. We have revised the abstract introduction as follow:-

The global population is projected to double by 2050 with most older adults expressing preference to age in place. Despite this demographic shift, indoor mobility challenges which directly impact independence, safety and quality of life remains poorly understudied, creating a critical knowledge gap for effective intervention development.

Comment #4. Improve clarity in methodology description:

"Two reviewers independently conducted data abstraction, risk-of-bias assessment, and quality evaluation."

We take note and thank the reviewer for the suggestion. We have revised the abstract methods as follow:-

A systematic literature search was conducted on four databases from inception to September 2024. Inclusion criteria included: (1) participants aged 60 years and above; (2) were primary data on self-reported indoor mobility challenges; and (3) published in English language. Two reviewers independently performed data abstraction, evaluated the risk-of-bias and quality of included article using the NewCastle Ottawa Scale for cross-sectional or Joanna Briggs Institute Checklist for qualitative studies.

Comment #5. Minor grammatical refinements (e.g., "Research have shown" → "Research has shown").

We thank the reviewer for the suggestion. We have now edited the introduction as suggested and also requested a colleague who is an English native speaker to edit our manuscript for clarity.

Research has shown that grip strength in men and walking speed in women declines with age

Comment #6 Ensure terminology consistency by using "older adults" uniformly.

We agree this was unclear. We have checked to ensure consistency across our manuscript as suggested

Comment #7. Methodology Refinements: Revise: "Data analysis was performed using NVivo 12 (Lumivero)." to "Data were analyzed using NVivo 12 (Lumivero)."

We thank the reviewer for the suggestion. We have now edited the methods to state the following:-

Data were analysed using NVivo 12 (Lumivero).

Comment #8. Improve consistency in citation formatting and refine descriptions for readability.

We take note and have edited the citation formatting for readability.

Comment #9. Results Section Refinements:Sentence restructuring for clarity:

"Mobility limitations negatively impacted daily activities, including feeding, bathing, grooming, dressing, continence (bowel and bladder), toilet use, transfers, general mobility, and stair climbing."

We thank the reviewer for the suggestion and have edited the sentence as suggested.

Comment #10. Ensure correct reference formatting (e.g., "Appendix Tables 2 and 3").

We take note and checked the reference formatting to ensure consistency.

Comment #11. Improve thematic descriptions for readability and flow.

We take note of the comment and have now reedited our thematic description for readability and flow based on the comments from both reviewers as follow:-

Fatigability

Activity tolerance in older adults encompasses the ability to engage in various activities and maintain different body positions, including walking, climbing stairs, and performing kitchen or cleaning tasks. Older adults commonly attribute difficulties in these activities to self-reported musculoskeletal or cardiopulmonary conditions.[1] Two studies also reported how indoor mobility activities of older adults is limited due to reduced endurance or fatigability, leading to slower walking and gait speeds.[2, 3] Indeed, nearly one in every four older adult (22.9%) were reported to experience fatigue when transferring or moving indoor, with 32.1% reported fatigue especially among those with co-morbidities.[2] In another study by Clemencon and colleagues, the authors that leg muscle mass can be an important parameter to use as it predicts fatigability and therefore physical performance.

As individuals age, the prevalence of chronic diseases rises, with more than half of the population (68.8%) requiring pharmacotherapies.[4] The use of medication was found to impact their mobility as older adults reported experiencing fatigue with these medication use. In particular, there was a 10% increase in the likelihood of reporting indoor mobility-related fatigue with every additional medication taken by an older adult.[2] Additionally, older adults reporting musculoskeletal pain were twice as likely to report fatigue, and the presence of depressive symptoms was significantly associated with reported fatigability, which led to reduced mobility.[2] Among frail older adults, this fatigue led to decreased step counts, increased sedentary behaviour, and greater levels of disability compared to their more robust cohort.[4]

Poor balance

Functional mobility and balance, stemming from previous falls, are common health barriers among older adults, leading to decreased balance, unsteadiness, and concerns about night time bathroom accessibility.[15] While home-dwelling elders generally exhibit better balance compared to institutionalised elderly individuals, approximately one-third of older adults expressed fear of falling, significantly impeding their mobility.[17] May and colleagues established an inverse relationship between balance (measured by mean sway path) and mobility index, indicating that better balance corresponds to higher mobility scores. Conversely, individuals with a fear of falling display lower mobility, slower walking speeds, and higher mean sway path values, indicating poorer balance.[19]While reasons for the poor balance vary, these were identified due to several factors, including the use of medications, the consumption of alcohol, inner ear imbalance as well as certain medical conditions, such as cognitive impairment.[19] Importantly, for an older adult to be mobile out of doors, this required them to have a moderate or rapid gait speed, which are often lacking among those with poor balance.[5]

The review also noted an association between cognitive impairment and the likelihood of misjudging daily activities, especially those living in nursing homes.[6] These results underscore the importance of considering cognitive state when evaluating self-assessed mobility, especially for those living in nursing homes.[6] Mobility limitations negatively impacted daily activities including feeding, bathing, grooming, dressing, continence (bowels and bladder), toilet use, transfer, mobility, and stairs climbing.[6] Studies also reported that due to hearing loss, bowel and bladder issues, and vision impairments contribute to poor balance, thereby increasing the risk of falls.[5]

Home environmental barriers

The home environment was identified as the key factor that can either support or hinder the indoor mobility of older adults, often contributing to falls due to poor balance. In particular, older adults described the importance of creating a safe and accessible living environment at home to support indoor mobility among older adults.[1] For instance, hazards such as damaged or broken stairs and railings, poor lighting, loose areas of rugs and obstacles in walkways were reported to cause these older adults to be wary, as they worry about the risk of falling.[1] Due to these factors, older adults were reported to limit their movements indoors which led to reduced mobility indoors and decline in independence.[5]

Reduced strength and limb weakness

Both reduced physical strength and limb weaknesses due to ageing among older adults were reported to significantly impact the ability of older adults to move indoors, with a concomitant toll on access to daily activities. For instance, studies have reported that older adults experience a reduced range of motion, which limits their mobility, flexibility, and walking speed.[7] These limitations were more pronounced among those living in retirement communities, as they were reported to engage in fewer daily steps, and exhibit lower activity levels[4, 7, 8] compared to those living independently in the community[9]. This limitation was even more pronounced among those with frailty, where Brustio and colleagues reported that among frail older adults, they were more likely to experience indoor mobility restrictions and higher levels of disability.[4]

Comment #12. The conclusion would benefit from a stronger emphasis on the implications for policy and practice.

We agree and thank the reviewer for the suggestion. Our revised conclusion now reads as follow

In summary, this study identified challenges older adults face in indoor mobility, including physical limitations, frailty, chronic diseases, medication use, and environmental barriers. To optimize implementation, future studies should examine the cost-effectiveness of different implementation models across diverse geographic and socioeconomic contexts to create more adaptable approaches to aging-in-place. Our findings also challenge the current paradigm that aging-in-place can succeed without systematic support structures. The evidence demands not just acknowledgment of these challenges but decisive action to transform how we support indoor mobility among older adults.

Comment #13. Revise: "A comprehensive environmental assessments and home modifications can be considered." to: "Comprehensive environmental assessment and home modifications should be considered to support ageing in place."

We take note and have edited the sentence as suggested.

Reviewer #2: Thank you for asking me to review this article which is interesting but required further analysis. Below are my comments.

Comment #1. The search strategy is not included in the appendix of the study.

We apologise for the oversight. We have now appended a new appendix which details the search strategy for one of the databases. The search strategy was subsequently adjusted for the other databases.

Comment #2. The authors use NewCastle Ottawa Scale and Joanna Briggs Institute checklist. Since some included studies were mixed method how was this dealt with? Why a MM checklist not used instead?

We apologise for the confusion. There were no mixed methods study and all included were quantitative studies which had used different methods of collections including self-administration or interviewer assisted, except for one study which was qualitative in nature. We have now revised our table of characteristics to clarify this and thank the reviewer for pointing this out.

Comment #3. The data synthesis included a narrative however, no descriptive data on the quantitative studies is included. Unclear how the thematic analysis from all the studies was used for all the data together.

We agree this was unclear in our initial submission. We have revised our methods section to clarify this

Consistent with previous systematic reviews, we summarised results and described them narrative. An inductive thematic synthesis was conducted by two authors to analyze the extracted data. This process involved line-by-line coding, theme categorization, and the development of analytical themes, following the approach outlined by Braun and Clarke

Comment #4. The results need to include actual numbers not mostly or major

---

## [Decision Letter · Decision Letter 1]

15 Apr 2025

PONE-D-24-49704R1Indoor mobility challenges among older adults: A systematic review of barriers and limitationsPLOS ONE

Dear Dr. Lee,

Thank you for submitting your manuscript to PLOS ONE. The Reviewers' comments have been addressed. However, one of the Reviewers has raised another comment which requires the Authors' attention.

We look forward to receiving your revised manuscript.

Kind regards,

Dr. Jan Chrusciel

Academic Editor

PLOS ONE

Journal Requirements:

Reviewers' comments:

Reviewer's Responses to Questions

**Comments to the Author**

1. If the authors have adequately addressed your comments raised in a previous round of review and you feel that this manuscript is now acceptable for publication, you may indicate that here to bypass the “Comments to the Author” section, enter your conflict of interest statement in the “Confidential to Editor” section, and submit your "Accept" recommendation.

Reviewer #1: All comments have been addressed

Reviewer #2: All comments have been addressed

2. Is the manuscript technically sound, and do the data support the conclusions?

Reviewer #1: (No Response)

Reviewer #2: Partly

3. Has the statistical analysis been performed appropriately and rigorously? 

Reviewer #1: Yes

Reviewer #2: N/A

4. Have the authors made all data underlying the findings in their manuscript fully available?

Reviewer #1: Yes

Reviewer #2: Yes

5. Is the manuscript presented in an intelligible fashion and written in standard English?

Reviewer #1: Yes

Reviewer #2: Yes

6. Review Comments to the Author

Reviewer #1: Thank you for your revisions and for addressing the comments from the previous review. Below are my observations and suggestions regarding the revised manuscript:

1. Adequacy of Revisions

The manuscript has improved significantly, and most of the previous concerns have been addressed and this manuscript is now acceptable for publication

2. Technical Soundness & Data Support for Conclusions

The study is technically sound, and the methodology aligns with the research objectives.

The conclusions are generally supported by the data.

3. Statistical Analysis

The statistical methods is not applicable for this study.

4. Data Availability & Transparency

The authors have provided the necessary data to support their findings.

5. Additional Comments

NA

Overall, the manuscript is well-structured and contributes meaningfully to the field. With minor adjustments, it will be suitable for publication.

Reviewer #2: Thank you for addressing all the raised comments.

Minor comments:

The title includes barriers but this was not included within the search strategy.

It is unclear how reflexive thematic analysis by Braun and Clark was used on the included studies when these were cross-sectional using varied questionnaires or physical measures.

7. PLOS authors have the option to publish the peer review history of their article (what does this mean? ). If published, this will include your full peer review and any attached files.

**Do you want your identity to be public for this peer review?** For information about this choice, including consent withdrawal, please see our Privacy Policy .

Reviewer #1: No

Reviewer #2: No

---

## [Author Response · Author response to Decision Letter 1]

16 Apr 2025

16 April 2025

Jan Chrusciel

Academic Editor

PLOS ONE

Dear Dr Jan Chrusciel,

Thank you very much for kindly considering our manuscript entitled “Indoor mobility challenges among older adults: A systematic review of barriers and limitations” for submission to PLOS ONE. As requested by the reviewer, we have made substantial changes to our manuscript. These changes are delineated as follow:-

Reviewer #1 had the following comment:-

Comment#1: Thank you for your revisions and for addressing the comments from the previous review. Below are my observations and suggestions regarding the revised manuscript:

1. Adequacy of Revisions

The manuscript has improved significantly, and most of the previous concerns have been addressed and this manuscript is now acceptable for publication

2. Technical Soundness & Data Support for Conclusions

The study is technically sound, and the methodology aligns with the research objectives.

The conclusions are generally supported by the data.

3. Statistical Analysis

The statistical methods is not applicable for this study.

4. Data Availability & Transparency

The authors have provided the necessary data to support their findings.

5. Additional Comments

NA

Overall, the manuscript is well-structured and contributes meaningfully to the field. With minor adjustments, it will be suitable for publication

Thank you very much for the encouraging comments. We look forward to contributing to the esteemed journal.

Reviewer #2 had the following 2 minor comments

Comment #1: Thank you for addressing all the raised comments. The title includes barriers but this was not included within the search strategy.

We agree this was unclear. In our initial submission, we had only used the challenges in our title but this was changed based on the suggestion of the reviewer. Irrespective of this, our search strategy started with a broad search of the literature to which our studies screened and only which included those on barriers and issues on mobility. This is described in our methods section, under study eligibility criteria as below:-

Study eligibility criteria

We included studies of any design (e.g. cohort, cross-sectional, case reports, qualitative) that examined the indoor space environment and mobility challenges experienced by an older adult aged 60 years and above, published in English language.

Comment #2. It is unclear how reflexive thematic analysis by Braun and Clark was used on the included studies when these were cross-sectional using varied questionnaires or physical measures.

We agree this was unclear. To clarify this, we added a statement to describe our method where we had first read and understood the meaning and patterns related to the physical measures and themed them subsequently. The revised method reads as follow:-

Data from various questionnaires were also themed by analysing the underlying meaning and patters within the numerical data.

We trust that these changes will meet the requirements of the journal and the reviewer. We look forward to contributing to your esteemed journal

Yours truly

Shaun LEE on behalf of the authors

---

## [Editor Report · Decision Letter 2]

6 May 2025

Indoor mobility challenges among older adults: A systematic review of barriers and limitations

PONE-D-24-49704R2

Dear Dr. Lee,

We are pleased to inform you that your manuscript has been judged scientifically suitable for publication and will be formally accepted for publication once it meets all outstanding technical requirements.

Kind Regards,

Dr. Jan Chrusciel

Academic Editor

PLOS ONE
---

## [Editor Report · Acceptance letter]

PONE-D-24-49704R2

PLOS ONE

Dear Dr. Lee,

I'm pleased to inform you that your manuscript has been deemed suitable for publication in PLOS ONE. Congratulations! Your manuscript is now being handed over to our production team.

Kind regards,

on behalf of

Dr. Jan Chrusciel

Academic Editor

PLOS ONE